# PESSIMISTIC REWARD MODELING IN RLHF AGAINST REWARD HACKING

## ABSTRACT

This work proposes 'PET', a novel pessimistic reward fine-tuning method, to learn a pessimistic reward model robust against reward hacking in offline reinforcement learning from human feedback (RLHF). Traditional reward modeling techniques in RLHF train an imperfect reward model, on which a KL regularization plays a pivotal role in mitigating reward hacking when optimizing a policy. Such an intuition-based method still suffers from reward hacking, and the policies with large KL divergence from the dataset distribution are excluded during learning. In contrast, we show that when optimizing a policy on a pessimistic reward model fine-tuned through PET, reward hacking can be prevented without relying on any regularization. We test our methods on the standard text generation datasets. We find that one can learn a high-quality policy on our pessimistic reward without using any regularization. **The learned policy has a high KL divergence from the dataset distribution while having high performance in practice. We also observe that the length bias phenomenon in reward modeling is significantly mitigated by PET.** While the proxy reward trained in traditional approaches shows bias to long responses, the pessimistic reward model finetuned by PET shows little bias. In summary, our work shows the feasibility of learning a pessimistic reward model through PET against reward hacking. The agent can greedily optimize a policy on the pessimistic reward without suffering from reward hacking. PET can also solve the length bias problem in reward modeling.

## 1 INTRODUCTION

Reinforcement learning from human feedback (RLHF) (Christiano et al., 2017) has become crucial in aligning large language models (LLMs), making LLMs more helpful, truthful, and harmless (Stiennon et al., 2020; Bai et al., 2022a; Ouyang et al., 2022; Rafailov et al., 2023). In a typical RLHF training framework (Ouyang et al., 2022), an agent first learns a reward model as a proxy for human preference that best interprets the preference training data. Then, the agent optimizes a policy on the proxy reward model to generate responses that achieve a high reward in expectation. In the real world, such a training framework faces the critical problem of 'reward hacking,' also known as 'reward over-optimization' (Eisenstein et al., 2023; Tien et al., 2022; Gao et al., 2023). Due to the dataset's size limitation, the proxy reward model is not always accurate. Given a prompt (task input), there could be a response (task output) that is not favored by human preference, but the learned proxy reward model overestimates the response and gives it a high reward. A greedy agent that searches for the policy with the highest proxy reward can learn to output such low-quality and overestimated responses. This is a typical instance of reward hacking. Due to the over-estimations in the proxy reward model, reward hacking can happen during policy optimization and induce undesired learning results (Gao et al., 2023), making it one of the most urgent challenges in RLHF.

**Challenges in preventing reward hacking:** To avoid reward hacking, the idea of 'pessimism' is necessary for the learning agent (Levine et al., 2020). At a high level, the agent must be pessimistic when evaluating the policy's performance based on the dataset. For example, in the framework described above, to avoid over-estimation, the agent should expect the case where the true performance of a policy is less than its performance evaluated based on the proxy reward. The key challenge here is controlling the degree of pessimism, that is, how low the true performance of a policy can be based on the preference dataset. While sufficient pessimism can avoid over-estimation and mitigate reward

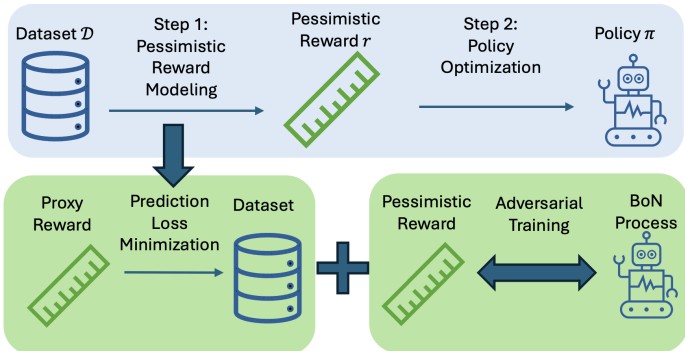

Figure 1: A novel two-step reward-based learning framework. The first step is a pessimistic reward modeling process. The reward model is trained to achieve a low loss in predicting the dataset preference while being pessimistic. More specifically, the pessimistic reward model is adversarially trained against a policy model induced by the BoN process such that the BoN policy achieves a low reward on the reward model. In the second step, the framework optimizes a policy on the pessimistic reward and outputs the learned policy.

hacking, over-pessimism results in an under-estimation of high-quality policies, causing the agent to exclude these good policies during training.

KL regularization plays a pivotal role in current RLHF methods to mitigate the risk of reward hacking (Ouyang et al., 2022). Intuitively, the KL divergence between the response distributions of a policy and the dataset can indicate the degree of uncertainty in evaluating the policy. A policy with higher uncertainty in evaluation requires the agent to be more pessimistic about it. Current algorithms explicitly (such as PPO (Schulman et al., 2017)) or implicitly (such as best-of-n sampling (Touvron et al., 2023)) rely on the KL divergence during policy optimization to mitigate the over-estimations in the proxy reward. However, such intuition-based methods are not always efficient in controlling pessimism. Reward hacking is still observed when using such methods (Gao et al., 2023; Rafailov et al., 2024), indicating that they are not pessimistic enough for certain policies. In the meantime, the methods may be overly pessimistic for some policies with a large KL divergence from the dataset distribution. This is verified by our empirical observations in Section 4. Recent studies use the adversarial training technique (Zhang et al., 2024; Xie et al., 2024; Liu et al., 2024b; Cen et al., 2024; Ji et al., 2024b) to achieve provable robustness against reward hacking. However, these methods are based on the direct policy optimization (DPO) technique (Rafailov et al., 2023) that requires KL regularization, which again unnecessarily induces over-pessimism. While KL regularization has almost always been considered necessary in practice, it is unlikely to be the most efficient approach to control pessimism. Therefore, a research question arises:

*Can efficient pessimism be achieved by learning a pessimistic reward model? Can an agent greedily optimize a policy on the pessimistic reward without using regularization to prevent reward hacking?*

**Best-of-n sampling with adversarial training:** In this work, we provide positive answers to the above questions. Our method is based on the adversarial training technique (Cheng et al., 2022; Bhardwaj et al., 2023; Zhan et al., 2023; Gupta et al., 2025) combined with best-of-n sampling (BoN) (Touvron et al., 2023; Beirami et al., 2024). In the framework, a policy model is trained to achieve a high reward on a reward model, while the reward model is trained to give a low expected reward to the responses generated by the policy model. Here, the main challenge comes from the complexity and instability of adversarial training. Intuitively, the policy model is optimized on different reward models during training, which complicates training convergence. Current works, such as Liu et al. (2024b), mitigate the challenge by using the DPO technique to simplify the training process. However, as discussed earlier, such a method requires KL regularization, which induces over-pessimism. To mitigate the challenge without using regularization, we consider using the BoN technique for the adversary against the reward model to simplify the adversarial training process. Given an initial policy model and a reward model, the BoN process samples several responses from the policy model and outputs the response with the highest reward as described in Alg 4 in the appendix. BoN performs policy optimization at inference time, making it easy to implement and compute. More importantly, we prove that **the adversarial training process with policies based on BoN can be solved by**

**a stochastic gradient descent process**, which is simple and computationally efficient. Note that traditional BoN suffers from reward hacking on the proxy reward (Touvron et al., 2023). Fortunately, the combination of BoN with adversarial training simultaneously solves the problems of reward hacking in BoN and the difficulty of policy optimization in adversarial training. Based on the idea of adversarial training against BoN, we develop a novel reward fine-tuning method and a corresponding two-step pessimistic RLHF algorithm. More specifically, our contributions are as follows:

1. We develop a novel reward fine-tuning method called 'PET'. A reward model fine-tuned by PET is pessimistic and robust against reward hacking. Under standard assumptions (Liu et al., 2024b), we theoretically prove that the BoN process on the pessimistic reward fine-tuned by PET has a performance comparable to the BoN process on any reward, as long as the corresponding policy induced by the process is covered by the dataset. While most RLHF algorithms are limited to the policies of low KL divergence from the initial policy, PET breaks the limit and learns high-performing policies with high KL divergence.

2. Based on 'PET', we develop a novel two-step pessimistic RLHF framework as shown in Fig. 1. The first step is learns a pessimistic reward model based on PET. The second step optimizes a policy on the learned pessimistic reward model. Note that the BoN process is only necessary for the first learning step. The second step can be achieved by policy optimization algorithms like PPO that require no BoN. In principle, the agent can greedily optimize a policy to achieve the highest reward on the pessimistic reward model without using any regularization. We empirically verify that the computational resource required by PET is limited.

3. We test our method on the summarization dataset (Völske et al., 2017b) and the IMDB dataset (Maas et al., 2011). On the summarization dataset, we observe that the BoN process on a pessimistic reward fine-tuned by PET significantly outperforms the traditional BoN process on the proxy reward. The win rate of the policy response against human response is increased from $32.0\%$ to $39.2\%$. We further use the PPO algorithm for policy optimization on the pessimistic reward, which we call 'PPO-PET'. We observe that the policy learned by PPO-PET has a high performance and a high KL divergence from the dataset distribution. This contradicts the traditional impression that policies with high KL divergence are vulnerable to reward hacking. On the summarization dataset, our policy in general has a comparable or higher performance compared to current state-of-the-art RLHF algorithms, including reward ensemble (RE) (Coste et al., 2023), DPO (Rafailov et al., 2023), RPO (Liu et al., 2024b), and $\chi$PO (Huang et al., 2024a). On the IMDB dataset, the PPO-PET method significantly outperforms other baselines.

4. We observe the notorious phenomenon of length bias on the proxy reward model trained with the traditional approach (Dong et al., 2024). The proxy reward model tends to give a higher reward for a longer response. We show that by finetuning the proxy reward model with PET, the learned pessimistic reward model exhibits much less bias on long responses. This suggests that our PET method can also be used to solve the length bias problem in reward modeling.

This work is closely related to the studies that design algorithms for RLHF, especially the algorithms robust against reward hacking. We provide a detailed discussion on related work in the appendix.

## 2 PRELIMINARIES

### 2.1 OFFLINE RLHF

In this work, we consider a standard RLHF problem in the offline learning setting. For a given task, let $\mathcal{X}$ be the input space (e.g., prompt), and $\mathcal{A}$ be the output space (e.g., response). There exists a preference model $\mathcal{P}(\cdot|\cdot,\cdot,\cdot) : \mathcal{X} \times \mathcal{A} \times \mathcal{A} \to \Delta(\{\succ, \prec\})$ that takes a prompt $x$ and a pair of responses $a_1, a_2$ as input and stochastically outputs a preference signal $\sigma \sim \mathcal{P}(\cdot|x, a_1, a_2)$, representing its preference on the two responses. For example, $\sigma$ being $\succ$ means the preference is $a_1 \succ a_2$ given the prompt $x$. In addition, we assume the preference model is a Bradley-Terry model (Bradley & Terry, 1952) associated with a reward model $r : \mathcal{X} \times \mathcal{A} \to \mathbb{R}$ as $\mathcal{P}_r(\succ |x, a_1, a_2) = \frac{1}{1+\exp\left(r(x,a_2)-r(x,a_1)\right)}$.

Here, the reward model $r$ represents the quality of a response. Given a prompt, a response with a higher reward is more likely to be preferred. We say $r$ is the true reward behind the preference model $\mathcal{P}_r$. A policy $\pi(\cdot|\cdot) : \mathcal{X} \to \Delta(\mathcal{A})$ takes a prompt $x$ as an input and stochastically outputs a response

$a \sim \pi(\cdot|x)$. Let $\mu$ be a probability distribution over the prompt space. The quality of the policy on the reward $r$ over the distribution $\mu$ is represented by $V_\mu^r(\pi) = \mathbb{E}_{x \sim \mu, a \sim \pi(\cdot|x)}[r(x, a)]$.

In the offline RLHF setting, there exists an offline preference dataset consisting of $N$ tuples $\mathcal{D} = \{(x^i, a_1^i, a_2^i, \sigma^i)\}_{i=1}^N$. The prompt and responses in each tuple is sampled from an i.i.d. distribution $(x^i, a_1^i, a_2^i) \sim \mu_\mathcal{D}$, and the preference signal is sampled from the preference model $\sigma^i \sim \mathcal{P}(\cdot|x^i, a_1^i, a_2^i)$. We call $\mu_D$ the dataset distribution and denote $\mu$ as the distribution of the prompts in $\mu_\mathcal{D}$. The agent has access to the offline dataset $\mathcal{D}$, the prompt space $\mathcal{X}$, the response space $\mathcal{A}$, and the prompt distribution $\mu$. Let $r^*$ be the true reward model behind the preference model that generates the preferences in the dataset. The general goal of the agent is to align a policy with the reward model reliably, that is, to find a policy of high performance $V_{r^*}^\mu(\pi)$ with a high probability.

## 2.2 Reward Modeling

Reward modeling is a process to learn a reward model that can well explain the preference signals in the dataset. Given a reward model $\hat{r}$, its prediction loss on the preference dataset $\mathcal{D}$ is defined as $\mathcal{L}_\mathcal{D}(\hat{r}) = \sum_{i=1}^N -\log \mathcal{P}_{\hat{r}}(\sigma^i|x^i, a_1^i, a_2^i)$. The prediction loss of a reward model represents its quality in interpreting the preference signals in the dataset. The process of traditional reward modeling is to find the reward model from a model class $\mathcal{R}$ (e.g., an LLM) that minimizes the prediction loss over the offline dataset: $\hat{r} \in \arg\min_{r \in \mathcal{R}} \mathcal{L}_\mathcal{D}(r)$. For convenience, we say a reward model is a 'proxy reward model' if it has low prediction loss on the dataset.

## 2.3 Policy Optimization

After learning a reward model, the next step is to optimize a policy on the reward model to achieve a high reward. In the appendix, we provide a detailed description of two typical policy optimization techniques most related to our study: KL regularized proximal policy optimization (KL-PPO) and best-of-n sampling (BoN). Here, we briefly go over BoN. BoN is an inference-time policy optimization method. Given a base policy model $\pi_0$, a reward model $\hat{r}$, and a positive integer $n$, the BoN process is defined in Alg 4 in the appendix. In practice, the reward model $\hat{r}$ is trained by minimizing the prediction loss, and the base policy $\pi_0$ is usually set as a proxy policy for the dataset. The proxy policy is trained to reproduce the response in the dataset. The BoN process is effectively a policy as it takes a prompt as input and stochastically outputs a response. We denote $\pi_{\text{BoN}}(\pi_0, \hat{r}, n)$ as the policy to represent the BoN process, which we call the 'BoN policy'. We highlight an intuitive yet important property of a BoN policy in Proposition 2.1. The proof is in the appendix. Later, we show that this property plays a critical role in our method.

**Proposition 2.1.** *For any prompt distribution $\mu$, base policy $\pi_0$, number of samples $n$, and reward model $r_0$, the BoN policies satisfy:*

$$V_{r_0}^\mu(\pi_{BoN}(\pi_0, r_0, n)) = \max_r V_{r_0}^\mu(\pi_{BoN}(\pi_0, r, n)).$$

## 3 Robust RLHF with Pessimistic Reward Finetuning (PET)

**Robust BoN as a minimax problem:** We achieve robustness against reward hacking by training a pessimistic reward model. Specifically, we aim to find a reward model with a low prediction loss on the dataset that gives a minimal relative score to its corresponding BoN policy. The relative score is compared to the score of a fixed reference policy, which we denote as $\pi_{\text{ref}}$. In practice, the reference policy is usually set as the proxy policy of the dataset. We focus on the relative score because the preference model only depends on the relative score between two responses.

Formally, to find such a pessimistic reward, we consider solving a minimax problem where a critic and an actor are trained against each other. The actor controls a policy model, and the critic controls a reward model. The minimax problem is as follows.

$$\min_{r \in \mathcal{R}} \max_{\pi \in \Pi_{\text{BoN}}^{n, \pi_0}} \left( V_r^\mu(\pi) - V_r^\mu(\pi_{\text{ref}}) \right) + \beta \cdot \mathcal{L}_\mathcal{D}(r). \tag{1}$$

Here, $\mathcal{R}$ is a reward model class, $\Pi_{\text{BoN}}^{n, \pi_0} = \{\pi_{\text{BoN}}(r, n, \pi_0) : r \in \mathcal{R}\}$ is the class of the BoN policies with base policy $\pi_0$, $n$ samples, and any reward model $r \in \mathcal{R}$. The minimax goal $f(\pi, r) :=$

---

**Algorithm 1** Pessimistic Reward Fine-Tuning (PET) with Adversarially Trained Best of N Sampling

---

1: **Input:** Initial reward model $\hat{r}$, Dataset $\mathcal{D}$, base policy $\pi_0$, reference policy $\pi_{\text{ref}}$, number of samples $n$, pessimistic coefficient $\beta$, learning rate $\alpha$
2: **Initialize:** $r^1 \leftarrow \hat{r}$
3: **for** $t = 1, \ldots, T$ **do**
4:     Update $\pi^t \leftarrow \pi_{\text{BoN}}(\pi_0, n, r^t)$
5:     Sample mini batch $\mathcal{D}_t = \{x_i, a_i^1, a_i^2, \sigma_i\}_{i=1}^M, (x_i, a_i^1, a_i^2, \sigma_i) \overset{\text{i.i.d}}{\sim} \mathcal{D}$
6:     Sample responses $a_i \sim \pi^t(\cdot|x_i), a_{\text{ref},i} \sim \pi_{\text{ref}}(\cdot|x_i), \forall i \in [M]$
7:     Compute pessimistic loss $l^t = \sum_{i \in [M]}[r^t(x_i, a_i) - r^t(x_i, a_{\text{ref},i}) + \beta \cdot \mathcal{L}_{\mathcal{D}_t}(r^t)]$
8:     Update $r^{t+1} \leftarrow r^t - \alpha \cdot \nabla l^t$
9: **end for**
10: **Return:** Reward model $r^{T+1}$

---

$\left(V_r^\mu(\pi) - V_r^\mu(\pi_{\text{ref}})\right) + \beta \cdot \mathcal{L}_{\mathcal{D}}(r)$ consists of two terms. The first term $\left(V_r^\mu(\pi) - V_r^\mu(\pi_{\text{ref}})\right)$ is a relative score between a policy $\pi$ and a reference policy $\pi_{\text{ref}}$ evaluated by the reward model $r$ over a prompt distribution $\mu$. The actor aims to find a policy that achieves a high score on the reward model, while the reward model aims to give a low relative score to the policy. The second term $\beta \cdot \mathcal{L}_{\mathcal{D}}(r)$ is the prediction loss on the preference dataset $\mathcal{D}$ multiplied by a positive weight $\beta > 0$, which is only determined by the reward model. This term constrains the reward model in that it must be a high-quality reward model that well explains the dataset.

Liu et al. (2024b) proposes to solve a similar minimax problem as ours. Here, we highlight two critical design in our framework, making it fundamentally different. The first design is that we do not consider KL regularization in our minimax problem, which is included in their work. Our insight is that the idea of pessimism is already included in formulating a minimax game between the policy and the reward models, so it is unnecessary to add KL regularization for more pessimism. In the appendix, we theoretically show that the KL regularization results in over-pessimism and decreases the learning efficiency. Our formulation has no regularization and achieves a tighter theoretical guarantee on learning efficiency. The second difference is that we focus on the policies based on BoN. BoN policies are easy to acquire and represent near-optimal policies on a reward model. More importantly, as we show next, the property of BoN in Proposition 2.1 makes it feasible to solve the minimax problem with stochastic gradient descent.

**PET and two-step pessimistic RLHF:** Based on the result from Proposition 2.1, we have $\pi_{\text{BoN}}(\pi_0, n, r) \in \max_{\pi \in \Pi_{\text{BoN}}^{n, \pi_0}} V_r^\mu(\pi)$. Therefore, the minimax problem in Eq. 1 can be simplified to a minimization problem as shown in Eq. 2.

$$\min_{r \in \mathcal{R}} \left(V_r^\mu(\pi_{\text{BoN}}(\pi_0, n, r)) - V_r^\mu(\pi_{\text{ref}})\right) + \beta \cdot \mathcal{L}_{\mathcal{D}}(r). \tag{2}$$

To solve the minimization above, we propose algorithm 'pessimistic reward fine-tuning' (PET) in Alg 1. In PET, the policy model and the reward model are iteratively updated. In each iteration, the policy model is updated through the BoN process on the current reward model. The reward loss is computed as the minimization goal in Eq. 2 with the updated policy model on a batch of prompts sampled from the dataset. Then, the reward model performs a gradient descent step on the reward loss. In the appendix, we prove that **PET is essentially a stochastic gradient descent process to solve the minimization goal in Eq. 2**, which is equivalent to solving the minimax problem in Eq. 1. This allows us to solve the minimax problem in a relatively simple approach.

PET proposes a pessimistic reward fine-tuning process to make a reward pessimistic. Based on PET, we develop a novel two-step pessimistic RLHF learning framework as shown in Alg 2. In this framework, the first step is a pessimistic reward modeling process. In practice, it first learns a proxy reward model that minimizes the prediction loss, and then fine-tunes the proxy reward with PET to make it pessimistic. In this setup, the PET training is initialized from the proxy reward with a low prediction loss, which intuitively reduces the complexity of pessimistic reward fine-tuning. The second step is a policy optimization process on the pessimistic reward. In principle, it is unnecessary in the last step to use additional pessimistic learning tricks such as regularization during policy optimization, as the reward model is already pessimistic. The agent can greedily optimize a policy to

---

**Algorithm 2** Two-Step Pessimistic RLHF Framework

1: **Input:** Dataset $\mathcal{D}$, Base policy $\pi_0$, Reference policy $\pi_{\text{ref}}$, number of samples $n$
2: **Step 1:** Traditional Reward modeling $\hat{r}_1 \leftarrow \min_{r \in \mathcal{R}} \mathcal{L}_{\mathcal{D}}(r)$ followed by pessimistic reward fine-tuning $\hat{r}_2 \leftarrow \textbf{PET}(\hat{r}_1)$
3: **Step 2:** Policy optimization $\hat{\pi} \leftarrow \max_{\pi \in \Pi} V_{\hat{r}_2}^{\mu}(\pi)$
4: **Return:** Policy model $\hat{\pi}$

---

achieve a high pessimistic reward without worrying about the risk of reward hacking. For example, in the standard KL-PPO algorithm, the optimization goal of the PPO algorithm is the reward of a policy regularized by the KL divergence between the policy and the proxy dataset policy. In our learning framework, the empirical results in Section 4 show that the PPO algorithm can learn a high-quality policy in reality on the pessimistic reward directly without using any regularization. This empirically verifies that the pessimistic reward fine-tuned by PET is trustworthy.

Compared to the traditional two-step learning framework, our learning framework adopts an additional PET process after the standard reward modeling process. This could require extra computational resources. In Appendix A.4, we provide a detailed analysis of PET's computational complexity. We highlight that in our experiments, the PET process only takes about $14\%$ of the total training time.

**Theoretical guarantees:** Here we provide theoretical guarantees on the optimality of the solution to Eq 1. The proof technique follows Liu et al. (2024b). We take a standard definition of the dataset coverage $C_{\mu}$ on a policy to characterize how well a dataset covers a policy $\pi$ in Definition A.2 in the appendix. Theorem 3.1 gives a theoretical guarantee on the performance of the policy in the solution to the minimax problem in Eq 1. The proof for Theorem 3.1 can be found in the appendix.

**Theorem 3.1.** *Consider a bounded reward model class $\mathcal{R}$ such that $\forall r \in \mathcal{R}, r(\cdot, \cdot) \in [-R, R]$ .Assume the true reward is included in the reward model class $r^* \in \mathcal{R}$. Let $\hat{\pi} = \pi_{BoN}(\pi_0, \hat{r}, n)$, where $\hat{r}$ is the reward model solution to the minimax problem in Eq 1. For any BoN policy $\pi \in \Pi_{BoN}^{n,\pi_0}$ covered by the dataset $\mathcal{C}_{\mu}(\pi, \pi_{ref}, \mathcal{R}) < +\infty$, by setting $\beta = \frac{\sqrt{N}(1+\exp(R))^2}{2\sqrt{6}\cdot\sqrt{\log(\frac{N_\epsilon(\mathcal{R},\|\cdot\|_\infty)}{\delta})}}$, with probability at least $1 - \delta$ the performance gap between $\hat{\pi}$ and $\pi$ is bounded by*

$$V_{r^*}^{\mu}(\pi) - V_{r^*}^{\mu}(\hat{\pi}) \leq \frac{(1 + \exp(R))^2 \cdot (\mathcal{C}_{\mu_{\mathcal{D}}}(\mathcal{R}, \pi, \pi_{ref})^2 + 1) \cdot \sqrt{6 \log(\frac{N_\epsilon(\mathcal{R},\|\cdot\|_\infty)}{\delta})}}{4\sqrt{N}},$$

*where $N_\epsilon(\mathcal{R}, \|\cdot\|_\infty)$ is the $\epsilon$-covering number of the reward model class (Cheng et al., 2022).*

*Remark* 3.2. It is standard in the related literature (Liu et al., 2024b; Zhan et al., 2023) to assume that the true reward is included in the reward model class. Theorem 3.1 shows that the performance of $\hat{\pi}$ is comparable to any BoN policy $\pi \in \Pi_{\text{BoN}}^{\pi_0,n}$ if the policy is covered by the dataset. As long as a high-quality policy is well covered by the dataset, the solution $\hat{\pi}$ will also have high quality, which indicates that reward hacking will not happen. This is an ideal result one can hope for in the offline learning setting (Huang et al., 2024a).

## 4 EXPERIMENTS

### 4.1 EXPERIMENT SETUP

**Dataset and model:** Our experiments focus on the standard TL;DR summarization datasets (Völske et al., 2017a; Stiennon et al., 2020) and use the Pythia-1b model (Biderman et al., 2023) for all policy and reward models during training. Here, the task is to generate a summary for a given piece of text. We consider the IMDB dataset (Maas et al., 2011) as an ablation study.

**Implementation:** For the reference policy $\pi_{\text{ref}}$, it is trained by the standard SFT process in Huang et al. (2024b). The base policy $\pi_0$ for the BoN process uses the same SFT model. During inference, the base policy $\pi_0$ uses a higher temperature ($T = 0.7$) than the reference policy ($T = 0.1$) to increase the randomness in the generation and boost the efficiency of BoN. Due to limitations on our computational resources, we set $n = 64$ for BoN during PET, while PET in principle can work with arbitrarily large values of $n$. The proxy reward model is trained by the standard reward modeling

process in Huang et al. (2024b). For fair comparison, all RLHF methods share the same reference policy and the proxy reward, if needed. We set the pessimistic coefficient $1/\beta = 0.1$ following the setups in Rafailov et al. (2023).

**Evaluation:** We evaluate the policy models learned by the RLHF algorithms on the test split of the TL;DR SFT dataset, which consists of {query, human summary} pairs. First, we use the policy model to generate a summary for each query in the dataset. Then, we use a judge LLM (Qwen-2.5-32B-Instruct) to evaluate the quality of the generated summaries against the human responses in the dataset. The win rate of the model against the human response is computed by the percentage of instances where the judge LLM preferred the model's summary over the human summary.

**Baselines:** We consider multiple state-of-the-art offline RLHF algorithms as baselines, including the traditional PPO-KL (Bai et al., 2022a), traditional BoN (Touvron et al., 2023) (on the proxy reward), DPO (Rafailov et al., 2023), reward ensemble (RE) (Coste et al., 2023), RPO (Liu et al., 2024b), and $\chi$-PO (Huang et al., 2024a). Our implementations of DPO and RPO are directly from the source code provided in the original papers. We follow the optimal setup reported by Coste et al. (2023) for the reward ensemble method. We re-implement the $\chi$-PO algorithm as the source code is unavailable. We use the same hyperparameter setup mentioned in the paper to ensure fair comparison. We try to re-implement another related RLHF baseline, 'InferenceTimePessimism' (Huang et al., 2025). The algorithm is based on the BoN process, and the number of samples is set as $n = 2^{13}$ in the paper. Due to our computational resource limitation, we set the number of samples as $n = 64$. This setup probably underestimates the method's efficiency, and its learned policy has a low win rate. Therefore, we skip this baseline.

## 4.2 Pessimistic Reward Finetuning Improves Best-of-N Sampling

Here, we evaluate our two-step learning framework with the BoN process for policy optimization. We test the number of samples to be $n = \{16, 32, 64, 128\}$ during PET and policy optimization, and the results are shown in Table 1. We observe that the BoN policy based on the pessimistic reward model significantly outperforms the one based on the proxy reward model. The finding empirically validates that the BoN process greatly benefits from the adversarial training process against a reward model.

|  | $n=16$ | $n=32$ | $n=64$ | $n=128$ |
|---|---|---|---|---|
| **BoN-PET** | **38.2** | **36.6** | **39.2** | **36.8** |
| BoN-Proxy | 28.4 | 32.0 | 32.0 | 34.2 |

Table 1: Win rate against human responses for the BoN process on the pessimistic or proxy reward with different numbers $n$ of samplings on the IMDB dataset. 'BoN-PET' means the BoN process is performed on the pessimistic reward fine-tuned by PET. Similarly, 'BoN-Proxy' means the reward model is the proxy reward. The results that achieve the highest win rates in each column are bolded.

## 4.3 Policy Optimization on Pessimistic Reward with No Regularization

|  | regularized | | **unregularized** | |
|---|---|---|---|---|
|  | win rate | KL | win rate | KL |
| **PPO-Pessimistic** | 36.0 | 11.6 | **40.8** | 114.0 |
| PPO-Proxy | **40.2** | 9.2 | 7.2 | 192.6 |

Table 2: The policies here are learned by the PPO algorithm on the pessimistic reward or proxy reward, with ('regularized') or without KL ('unregularized') regularization. We show the win rate against human responses of the policies and their KL divergence to the reference policy. The win rates that are the highest ones in each column are bolded.

As introduced earlier, the proxy reward model may overestimate the rewards for out-of-distribution prompt responses, so KL regularization is always considered necessary during policy optimization to prevent reward hacking. In this section, we show that our pessimistic reward model, which is fine-tuned on the proxy reward, is robust against reward hacking such that no regularization is necessary when optimizing a policy on the pessimistic reward. Here, we use the PPO algorithm for the policy optimization step in the two-step pessimistic learning framework. We compare the

two cases where KL regularization or no regularization is applied during policy optimization, and we train the algorithms on the proxy reward and the pessimistic reward. Without regularization, the agent would greedily search for the policy with the highest reward. Such an agent is usually considered vulnerable to reward hacking. The results are shown in Table 2. We observe that for the proxy reward model, the policy learned under the KL regularization performs much better than one with no regularization. The results confirm that the proxy reward model is not pessimistic, and greedily optimizing a policy on proxy reward without KL regularization is vulnerable to reward hacking. In contrast, for the pessimistic reward model, the policy learned under no regularization performs better than the one with regularization. The results intuitively show that the pessimistic fine-tuning successfully makes the reward model pessimistic, and a greedy agent is robust against reward hacking when learning from the pessimistic reward model without using any regularization.

We notice another important finding from Table 2. Traditionally, the experience is that policies with high KL divergence from the reference policy have a high risk of reward hacking, so they should be excluded during learning through regularization. This agrees with the case of optimizing a policy on the proxy reward with no regularization, as the learned policy has low performance and high KL divergence. However, such an experience does not hold for the case when the reward model is a pessimistic reward. We find that the policy learned on the pessimistic reward fine-tuned by PET has a high performance in reality while having a large KL divergence. This is a sign of over-pessimism when using KL regularization for pessimism, as such good policies are excluded. Our empirical results prove that it is possible to find a high-quality policy on the dataset with a large KL divergence from the reference policy. To include such policies during learning without causing reward hacking, one can perform policy optimization on the pessimistic reward fine-tuned by PET.

In the appendix, we provide a detailed discussion on the computational resources and complexity required by our learning framework. We highlight that the PET process in our experiments only takes 14.3% of the total training time of the two-step learning framework.

## 4.4 LENGTH BIAS OF REWARD MODELS

Here, we empirically investigate the length bias phenomenon in our learning framework. It is a known issue that the proxy reward trained on the preference dataset is biased towards long responses (Dong et al., 2024; Singhal et al., 2023). To evaluate the length bias of the reward models, we sample 500 prompts from the dataset. For each prompt, we generate a response from the SFT model trained on the dataset. Then, we score each prompt-response with the reward model to evaluate. Figure 2 shows a scatter plot of the length of response versus the score given by the reward models. We find that the proxy reward trained by the traditional reward modeling approach is clearly inclined to give a high reward for the response of a long length. This is aligned with the aforementioned length bias phenomenon. In contrast, when we finetune the proxy reward through PET, the learned pessimistic reward is much less biased towards long responses. Note that our training framework never uses the length of responses for training. The empirical results suggest that PET is a promising method to solve the length bias issue in reward modeling. An explanation for the result is that PET trains a reward model to be pessimistic, which never overestimates. Since the length bias is a kind of over-estimation, the pessimistic reward model trained by PET should also avoid this kind of over-estimation.

## 4.5 COMPARISON BETWEEN ALGORITHMS ON SUMMARIZATION AND IMDB DATASETS

Here, we compare our method against the RLHF baselines. We consider using the PPO algorithm for policy optimization with no KL regularization in our two-step learning framework, and we call it 'PET-PPO'. Since the RLHF methods we consider generally apply to other NLP tasks in principle, we evaluate them on the IMDB dataset as well to make the evaluation more comprehensive. In this task, the model is prompted with a prefix to a movie review, and it must generate a continuation of that review with positive sentiment. Our data construction and evaluation metric exactly follows the setups in Rafailov et al. (2023). To make the task more challenging, the size of our dataset is about 25% of the dataset's size described in Rafailov et al. (2023). Training with less data can better highlight the sampling efficiency of an algorithm. The evaluation metric is the likelihood of positive sentiment for the completions generated by the model judged by an LLM. The same LLM is used to generate the preference labels in the dataset, so the likelihood is the ground truth in the task.

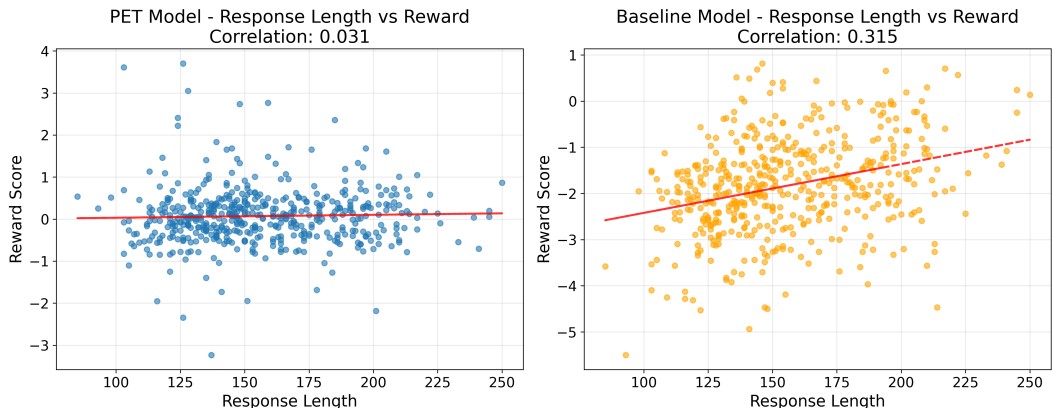

Figure 2: The scatter plot of the reward model's prediction on responses of various lengths. The baseline model is the proxy reward trained through the traditional approach that minimizes the prediction loss. The PET model is the pessimistic reward finetuned from the proxy reward by the PET algorithm.

|  | SFT | BoN-proxy | KL-PPO | RE | DPO | RPO | $\chi$PO | **PET-PPO** |
|---|---|---|---|---|---|---|---|---|
| Summarization | 25.6 | 32.0 | **40.2** | **40.2** | 38.8 | 34.0 | 39.2 | **40.8** |
| IMDB | 89.6 | 90.7 | 91.8 | 93.5 | 95.0 | 93.5 | 92.8 | **98.3** |

Table 3: Performance of different RLHF methods learning on the summarization and IMDB datasets. The metric on the summarization dataset is the win rate against human responses. On the IMDB dataset, the metric is the likelihood of the responses to have positive sentiment. We bold the results that achieve the highest score on the metric or a score no less than the highest score by at most 1.

Table 3 shows the standard evaluations on the performance of the policies learned by different algorithms. On the IMDB dataset, our method significantly outperforms the baselines by a clear margin. Note that in the IMDB dataset, a policy model is strictly better than another model if it achieves a higher evaluation score. On the summarization dataset, our method outperforms other methods in general. Here, the win rate is compared against human response, so a model does not necessarily beat another model if it has a higher win rate. Therefore, we directly compare policy models learned by the baselines against our policy model. In Table 4, we show the win rate of our PPO-PET method against other methods. We observe that, in general, the policy model learned by PPO-PET has a win rate above 50% against other models. The empirical results show that the performance of our PET-PPO method is very competitive compared to current SOTA RLHF methods.

|  | SFT | BoN-proxy | KL-PPO | RE | DPO | RPO | $\chi$PO |
|---|---|---|---|---|---|---|---|
| **PET-PPO** | **72.0** | **61.6** | **57.2** | 50.4 | 50.1 | **58.4** | 52.8 |

Table 4: Direct comparison between our PET-PPO method against other RLHF baselines on the summarization dataset. We bold the win rates that are above 55%.

## 5 Conclusions and Limitations

This work develops PET, a reward fine-tuning method. A pessimistic reward model can be learned by fine-tuning a proxy reward with PET. The pessimistic model shows limited bias towards long responses. When optimizing a policy on a pessimistic reward, a greedy agent can learn a high-performing policy with no regularization. We develop an RLHF method called PET-PPO that uses the PPO algorithm for policy optimization on a pessimistic reward fine-tuned by PET. Our empirical results show that PET-PPO achieves comparable or stronger performance on the IMDB and the TL;DR summarization dataset than multiple current SOTA RLHF baselines. The scope of this work is limited to RLHF in the offline setting. Due to limited computational resources, the evaluation is performed on a subset of RLHF tasks based on LLMs with relatively small size.

## 6 ETHICS STATEMENT

This paper studies an RLHF learning algorithm whose goal is to advance the field of Machine Learning. After carefully reading the code of ethics, we believe there are no ethical concerns about this work.

## 7 REPRODUCIBILITY STATEMENT

For the theoretical statements in Section 3, a clear explanation for the assumptions is provided in the same section, and a detailed proof is provided in the appendix. For the empirical results in Section 4, a clear explanation for the basic experiment setup, such as the datasets and models, is provided in the same section. A detailed setup for the hyperparameters, such as the learning rate and batch size, is provided in the appendix. The source code to reproduce the results is provided in the supplementary materials.

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

# A  APPENDIX

## A.1  DETAILED THEORETICAL ANALYSIS

### A.1.1  PET SOLVES EQ. 1 WITH A STOCHASTIC GRADIENT DESCENT PROCESS

Here, we explain in detail how the PET algorithm in Alg 1 solves the minimax problem in Eq. 1 in a stochastic gradient descent approach. Recall that the minimax problem follows:

$$\min_{r \in \mathcal{R}} \max_{\pi \in \Pi_{\mathrm{BoN}}^{n, \pi_0}} \left( V_r^\mu(\pi) - V_r^\mu(\pi_{\mathrm{ref}}) \right) + \beta \cdot \mathcal{L}_\mathcal{D}(r).$$

By Proposition 2.1, we have $\pi_{\mathrm{BoN}}(\pi_0, n, r^t) \in \arg\max_{\pi \in \Pi_{\mathrm{BoN}}^{n, \pi_0}} V_r^\mu(\pi)$. Therefore, solving the minimax problem in Eq 1 is equivalent to solving the minimization problem as follows:

$$\min_{r \in \mathcal{R}} \big( V_r^{\mu}(\pi_{\text{BoN}}(\pi_0, n, r)) - V_r^{\mu}(\pi_{\text{ref}}) \big) + \beta \cdot \mathcal{L}_{\mathcal{D}}(r).$$

Denote

$$f(r, \pi) := \big( V_r^{\mu}(\pi) - V_r^{\mu}(\pi_{\text{ref}}) \big) + \beta \cdot \mathcal{L}_{\mathcal{D}}(r)$$

and

$$h(r) := \big( V_r^{\mu}(\pi_{\text{BoN}}(\pi_0, n, r^t)) - V_r^{\mu}(\pi_{\text{ref}}) \big) + \beta \cdot \mathcal{L}_{\mathcal{D}}(r).$$

Then, our goal is to solve the problem $\min_{r \in \mathcal{R}} h(r)$. For any reward model $r_0 \in \mathcal{R}$, we have

$$\nabla h(r)|_{r=r_0} = \nabla_r f(r, \pi)|_{r=r_0, \pi=\pi_{\text{BoN}}(\pi_0, n, r_0)} + \nabla_\pi f(r, \pi)|_{r=r_0, \pi=\pi_{\text{BoN}}(\pi_0, n, r_0)} \cdot \nabla_r \pi_{\text{BoN}}(\pi_0, n, r_0)|_{r=r_0}$$

$$= \nabla_r f(r, \pi)|_{r=r_0, \pi=\pi_{\text{BoN}}(\pi_0, n, r_0)}$$

The second equality holds because $\pi = \pi_{\text{BoN}}(\pi_0, n, r)$ is the optimizer for $\max_{\pi \in \Pi_{\text{BoN}}^{n, \pi_0}} f(r, \pi)$, so we have $\nabla_\pi f(r, \pi)|_{r=r_0, \pi=\pi_{\text{BoN}}(\pi_0, n, r_0)} \equiv 0, \forall r_0 \in \mathcal{R}$. This is an important result as it shows that to compute $\nabla h(r)|_{r=r_0}$, we don't need to compute $\nabla_r \pi_{\text{BoN}}(\pi_0, n, r)|_{r=r_0}$. The latter represents how the BoN policy would change if the reward function changes, which is hard to approximate in practice.

A standard approach to find the minimizer of $\min_r h(r)$ is to perform stochastic gradient descent on $h(r)$. By the previous results, we can implement this process by approximating the value of $\nabla h(r)|_{r=r_0} = \nabla_r f(r, \pi)|_{\pi=\pi_{\text{BoN}}(\pi_0, n, r^t))}$.

Next, we show that PET in Alg 1 is essentially the process of stochastic gradient descent to solve the problem $\min_{r \in \mathcal{R}} h(r)$. Recall $V_\mu^r(\pi) = \mathbb{E}_{x \sim \mu, a \sim \pi(\cdot|x)}[r(x, a)]$. At line 5, the prompts $x_i$ and the data minibatch $\mathcal{D}_t$ are sampled from the dataset $\mathcal{D}$. At line 6, the responses $a_i, a_{\text{ref},i}$ are stochastically sampled from the current policy $\pi^t$ and the reference policy $\pi_{\text{ref}}$. Therefore, the expectation of the average loss $l^t/M$ equals to the value of $h(r^t)$:

$$\mathbb{E}\big[ \sum_{i \in [M]} [r^t(x_i, \pi^t(x^i, a_i)) - r^t(x_i, \pi^t(x^i, a_{\text{ref},i})) + \beta \cdot \mathcal{L}_{\mathcal{D}_t}(r^t)]\big]/M = \big( V_r^{\mu}(\pi^t) - V_r^{\mu}(\pi_{\text{ref}}) \big) + \beta \cdot \mathcal{L}_{\mathcal{D}}(r)$$

At line 8, the algorithm computes the gradient of $l^t$, so Alg 1 is essentially a standard process of stochastic gradient descent to solve the minimization problem $\min_{r \in \mathcal{R}} h(r)$. Therefore, PET is a convenient implementation to solve the minimax problem based on BoN in Eq. 1.

*Remark* A.1. Here, we argue that PET approximates the most pessimistic low-prediction-loss reward model against the BoN process. Denote $\hat{r}$ the solution to the minimization problem $\hat{r} \in \min_{r \in \mathcal{R}} \big( V_r^{\mu}(\pi) - V_r^{\mu}(\pi_{\text{ref}}) \big) + \beta \cdot \mathcal{L}_{\mathcal{D}}(r)$. For any reward model $r \in \mathcal{R}$ that has the same or lower prediction loss on the dataset $\mathcal{L}_{\mathcal{D}}(r) \leq \mathcal{L}_{\mathcal{D}}(\hat{r})$, we have $\big( V_{\hat{r}}^{\mu}(\pi) - V_{\hat{r}}^{\mu}(\pi_{\text{ref}}) \big) \leq \big( V_r^{\mu}(\pi) - V_r^{\mu}(\pi_{\text{ref}}) \big)$. This is because

$$\big( V_{\hat{r}}^{\mu}(\pi) - V_{\hat{r}}^{\mu}(\pi_{\text{ref}}) \big) + \beta \cdot \mathcal{L}_{\mathcal{D}}(r) \leq \big( V_r^{\mu}(\pi) - V_r^{\mu}(\pi_{\text{ref}}) \big) + \beta \cdot \mathcal{L}_{\mathcal{D}}(r)$$

$$\Rightarrow \big( V_{\hat{r}}^{\mu}(\pi) - V_{\hat{r}}^{\mu}(\pi_{\text{ref}}) \big) - \big( V_r^{\mu}(\pi) - V_r^{\mu}(\pi_{\text{ref}}) \big) \leq \beta \cdot (\mathcal{L}_{\mathcal{D}}(r) - \mathcal{L}_{\mathcal{D}}(\hat{r}))$$

$$\Rightarrow \big( V_{\hat{r}}^{\mu}(\pi) - V_{\hat{r}}^{\mu}(\pi_{\text{ref}}) \big) - \big( V_r^{\mu}(\pi) - V_r^{\mu}(\pi_{\text{ref}}) \big) \leq 0$$

$$\Rightarrow \big( V_{\hat{r}}^{\mu}(\pi) - V_{\hat{r}}^{\mu}(\pi_{\text{ref}}) \big) \leq \big( V_r^{\mu}(\pi) - V_r^{\mu}(\pi_{\text{ref}}) \big)$$

In practice, we use the PET algorithm to approximate the solution $\pi_0$ and find that the prediction loss of the learned reward model is as low as that of the proxy reward, which is specially trained to minimize the prediction loss. This implies that the reward model $\hat{r}$ is also the most pessimistic reward model that gives a minimal relative score to the BoN process among the reward models with low values of prediction loss. This observation also supports our argument that the reward model learned from PET is pessimistic.

### A.1.2 PROOF FOR PROPOSITION 2.1

Proposition 2.1 describes an intuitive property of BoN policies. It is equivalent to the statement that when optimizing a policy on a reward model $r_1$ through the BoN process, the BoN process that sets its reward model as $r = r_1$ achieves the highest reward on $r_1$. For any BoN process with the same base policy and number of samples, the distribution of the sampled responses during the process is always the same. Therefore, for any outcome of sampled responses, the BoN process that sets $r = r_1$ can always output the response with the highest reward on $r_1$. Formally, the proof for Proposition 2.1 is as follows.

*Proof.* Given a base policy $\pi_0$, a positive integer $n$, and a prompt $x$, consider a stochastic process where $n$ responses are i.i.d drawn from the policy at the prompt $a_i \overset{i.i.d}{\sim} \pi_0(\cdot|x), i \in [n]$. Let $E(x)$ be the space of all possible outcomes. For any outcome $e \in E(x)$, let $\{a_1, \dots, a_n\}$ be the sampled responses. For any two different reward models $r_1 \neq r_2$, denote $i_1^* \in \arg\max_{i \in [n]} r_1(x, a_i)$ and $i_2^* \in \arg\max_{i \in [n]} r_2(x, a_i)$ as the optimal index on the two rewards. Denote $v_1(e, r) := r(x, a_{i_1^*})$ and $v_2(e, r) := r(x, a_{i_2^*})$ where $r$ is any reward model, then by definition, we have $v_1(e, r_1) \geq v_2(e, r_1)$. Consider two BoN policies $\pi_1 = \pi_{\text{BoN}}(\pi_0, r_1, n), \pi_2 = \pi_{\text{BoN}}(\pi_0, r_2, n)$ defined on the base policy $\pi_0$, sampling number $n$, and the reward models $r_1, r_2$. Their performance on the reward model $r_1$ for any prompt distribution $\mu$ satisfies $V_{r_1}^\mu(\pi_1) = \mathbb{E}_{x \sim \mu, e \sim E(x)}[v_1(e, r_1)]$, $V_{r_1}^\mu(\pi_2) = \mathbb{E}_{x \sim \mu, e \sim E(x)}[v_2(e, r_1)]$. By the previous result $v_1(e, r_1) \geq v_2(e, r_1)$, we have $V_{r_1}^\mu(\pi_1) \geq V_{r_1}^\mu(\pi_2)$, which conclude the proof.

$\square$

### A.1.3 PROOF FOR THEOREM 3.1

First, we present the definition of the coefficient of dataset coverage on a policy.

**Definition A.2.** For a policy $\pi$, given a reference policy $\pi_{\text{ref}}$, a dataset distribution $\mu_\mathcal{D}$, and a reward model class $\mathcal{R}$, the coverage coefficient is defined as Zhan et al. (2023) $\mathcal{C}_{\mu_\mathcal{D}}(\mathcal{R}, \pi, \pi_{\text{ref}}) :=$

$$\max\left\{0, \sup_{r \in \mathcal{R}} \frac{\mathbb{E}_{x \sim \mu, a_1 \sim \pi(\cdot|x), a_2 \sim \pi_{\text{ref}}(\cdot|x)}\left[\left(r^*(x, a_1) - r^*(x, a_2)\right) - \left(r(x, a_1) - r(x, a_2)\right)\right]}{\mathbb{E}_{x, a_1, a_2 \sim \mu_\mathcal{D}}\left[|\left(r^*(x, a_1) - r^*(x, a_2)\right) - \left(r(x, a_1) - r(x, a_2)\right)|^2\right]}\right\},$$

where $\mu$ is the prompt distribution in $\mu_\mathcal{D}$.

Next, we provide a detailed proof for Theorem 3.1.

*Proof.* Our proof generally follows the proof technique in Liu et al. (2024b). Let $\hat{r}$ be the reward solution to $\hat{r} \in \arg\min_{r \in \mathcal{R}}$

$V_{r^*}^\mu(\pi) - V_{r^*}^\mu(\hat{\pi})$

$= \left(V_{r^*}^\mu(\pi) - V_{\hat{r}}^\mu(\pi)\right) + \left(V_{\hat{r}}^\mu(\pi) - V_{\hat{r}}^\mu(\hat{\pi})\right) + \left(V_{\hat{r}}^\mu(\hat{\pi}) - V_{r^*}^\mu(\hat{\pi})\right)$

$\leq \left(V_{r^*}^\mu(\pi) - V_{\hat{r}}^\mu(\pi)\right) + \left(V_{\hat{r}}^\mu(\hat{\pi}) - V_{r^*}^\mu(\hat{\pi})\right)$

$= \left(V_{r^*}^\mu(\pi) - V_{\hat{r}}^\mu(\pi)\right) + \left(V_{\hat{r}}^\mu(\hat{\pi}) - V_{\hat{r}}^\mu(\pi_{\text{ref}}) + \beta \cdot \mathcal{L}_\mathcal{D}(\hat{r}) - (V_{r^*}^\mu(\hat{\pi}) - V_{\hat{r}}^\mu(\pi_{\text{ref}}) + \beta \cdot \mathcal{L}_\mathcal{D}(r^*))\right) +$

$\beta \cdot \left(\mathcal{L}_\mathcal{D}(r^*) - \mathcal{L}_\mathcal{D}(\hat{r})\right)$

$\leq \left(V_{r^*}^\mu(\pi) - V_{\hat{r}}^\mu(\pi)\right) + \beta \cdot \left(\mathcal{L}_\mathcal{D}(r^*) - \mathcal{L}_\mathcal{D}(\hat{r})\right)$

The first equality utilizes the optimality of the policy solution $\hat{\pi} \in \arg\max_{\pi \in \Pi_{\text{BoN}}} V_{\hat{r}}^\mu(\pi)$. The second equality utilizes the optimality of the reward solution $\hat{r} \in \arg\min_{r \in \mathcal{R}} \left(V_{\hat{r}}^\mu(\pi) - V_{\hat{r}}^\mu(\pi_{\text{ref}}) + \beta \cdot \mathcal{L}_\mathcal{D}(r)\right)$. Intuitively, the formulation in the last line is bounded for any reward function $\hat{r}$. If the reward function is close to the true reward $r^*$, then both differences $V_{r^*}^\mu(\pi) - V_{\hat{r}}^\mu(\pi)$ and $\mathcal{L}_\mathcal{D}(r^*) - \mathcal{L}_\mathcal{D}(\hat{r})$ should be small. If $\hat{r}$ is very different from $r^*$, then its prediction loss on the dataset $\mathcal{L}_\mathcal{D}(\hat{r})$ would also be high, so that the performance gap can still be bounded. This indicates the importance of adding the prediction loss as a constraint on the reward model in Eq. 1. Formally, based on the results of Theorem 5.3 in Liu et al. (2024b), with probability at least $1 - \delta$, the last term can be bound by

$$V_{r^*}^{\mu}(\pi) - V_{r^*}^{\mu}(\hat{\pi}) \leq \frac{\mathcal{C}_{\mu_{\mathcal{D}}}(\mathcal{R}, \pi, \pi_{\text{ref}})^2}{8\kappa^2 \cdot \beta} + \frac{3\beta}{N} \log(\frac{N_{\epsilon}(\mathcal{R}, \|\cdot\|_{\infty})}{\delta}).$$

Here, $\kappa = \frac{1}{(1+\exp(R))^2}$ is a constant, and $N_{\epsilon}(\mathcal{R}, \|\cdot\|_{\infty})$ is the $\epsilon$-covering number for the reward model class Cheng et al. (2022). Setting

$$\beta = \frac{\sqrt{N}}{2\kappa \cdot \sqrt{6\log(\frac{N_{\epsilon}(\mathcal{R}, \|\cdot\|_{\infty})}{\delta})}}$$

concludes the proof. Note that the bound here is tighter than the bound on Liu et al. (2024b), because there is no KL regularization in our formulation at Eq. 1. Adding a KL regularization multiplied by any positive coefficient to Eq. 1 will increase the bound on the performance gap analysis, which is undesired for sampling efficiency. □

## A.2 Policy Optimization Methods

**KL regularized proximal policy optimization (KL-PPO):** For a prompt distribution $\mu$, the 'KL divergence' between two policies can be defined as $\text{KL}_{\mu}(\pi_1, \pi_2) := \mathbb{E}_{x\sim\mu}[\text{KL}(\pi_1(\cdot|x)\|\pi_2(\cdot|x))]$, where $\text{KL}(\pi_1(\cdot|x)\|\pi_2(\cdot|x)) := \sum_{a\in\mathcal{A}} \pi_1(a|x) \cdot \log \frac{\pi_1(a|x)}{\pi_2(a|x)}$ is the standard KL divergence between two distributions. Consider a proxy policy $\pi_{\text{ref}}$ of the dataset. The proxy policy is trained to generate a response distribution that is similar to the dataset response distribution. This is often achieved by a standard supervised fine-tuning process on the prompts and responses from the dataset $\mathcal{D}$ (Touvron et al., 2023). The KL divergence between a policy $\pi$ and $\pi_{\text{ref}}$ intuitively indicates how well it is covered by the dataset. Then, the learning goal of the agent becomes

$$\hat{\pi} \leftarrow \arg\max_{\pi\in\Pi} V_{\hat{r}}^{\mu}(\pi) + \eta \cdot \text{KL}_{\mu}(\pi, \pi_{\text{ref}}),$$

where $\Pi$ is a model family, $\eta > 0$ is the weight of the KL regularization in the optimization target. The most popular way to solve this optimization problem is by using the PPO algorithm (Ouyang et al., 2022). Combining the reward modeling step and the policy optimization process gives the popular RLHF algorithm 'PPO-KL' as shown in Alg 3

**Best-of-N sampling process (BoN):** Best-of-N sampling is an inference-time policy optimization method. Given a base policy model $\pi_0$, a reward model $\hat{r}$, and a positive integer $n$, the process of BoN is defined in Alg 4. In practice, the reward model $\hat{r}$ is trained by minimizing the prediction loss, and the base policy $\pi_0$ is usually set as a proxy policy for the dataset. The BoN process is effectively a policy as it takes a prompt as input and stochastically outputs a response. A BoN policy can be directly implemented on a policy model and a reward model without any training. It also achieves a higher reward on the reward model $\hat{r}$ compared to the base policy $\pi_0$. The pessimism in BoN implicitly relies on the notion of KL regularization. When the value of $n$ is small, the BoN policies will have a limited KL regularization compared to the base policy $\pi_0$, which intuitively reduces the risk of reward hacking Beirami et al. (2024). One can achieve a higher performance on the reward model with BoN by increasing the value of $n$, but in this case, the risk of reward hacking also increases Huang et al. (2025). In practice, reward hacking has been observed in BoN with a relatively high value of $n$ Touvron et al. (2023); Gao et al. (2023). Therefore, it is important to develop principled methods to free BoN policies from reward hacking.

---

**Algorithm 3** KL-PPO

1: **Input:** Reference policy $\pi_{\text{ref}}$, Dataset $\mathcal{D}$, KL weight $\eta$
2: **Step 1:** Reward modeling $\hat{r} \leftarrow \min_{r\in\mathcal{R}} \mathcal{L}_{\mathcal{D}}(r)$
3: **Step 2:** Policy optimization with PPO $\hat{\pi} \leftarrow \max_{\pi\in\Pi} V_{\hat{r}}^{\mu}(\pi) - \eta \cdot \text{KL}_{\mu}(\pi, \pi_{\text{ref}})$
4: **Return:** Policy model $\hat{\pi}$

---

## A.3 Additional Experiment Details

**Hyper-parameter setups:**

---

**Algorithm 4** Best-of-n sampling process

---

1: **Input:** Base policy $\pi_0$, reward model $\hat{r}$, number of samples $n$, prompt $x$
2: Draw responses $a_i \overset{i.i.d.}{\sim} \pi_0(\cdot|x), \forall i \in [n]$
3: Generate rewards $r_i = \hat{r}(x, a_i), \forall i \in [n]$
4: **Return:** Response $a_{i^*}$, where $i^* \in \arg\max_{i \in [n]} r_i$

---

| Configuration | PET | PPO |
|---|---|---|
| learning rate | $3e-8$ | $3e-6$ |
| learning scheduler type | cosine | cosine |
| batch size | 128 | 128 |
| gradient accumulation steps | 16 | 16 |
| training epoch | 1 | 1 |
| pessimistic coefficient $\beta$ | 10 | / |
| KL regularization weight $\eta$ | / | 0.05 |
| optimizer | adamw torch | adamw torch |
| precision | bfloat16 | bfloat16 |

Table 5: Training configurations for PET and PPO.

We list detailed training configurations for PET and PPO in Table 5. For standard SFT model and proxy reward model training, we follow the same setup as in (Huang et al., 2024b). For all baseline methods, we follow the original authors' implementations or descriptions. Specifically, for RPO, we remove the default chat template provided by its source code, as it is unsuitable for our summarization and IMDB tasks. This is confirmed by our observation that including the template degrades RPO's performance on our tasks, so the template is not included in our RPO training.

### A.4 COMPUTATIONAL COST AND COMPLEXITY

**Computational Resources:** First, we show the computational resource required by our training framework, especially the PET step that trains a pessimistic reward model. Our experiments are conducted using an A100 GPU with 40GB of VRAM and a 48-core Intel Xeon Silver 4214R CPU. Here, we show the details of the running time of our experiments on the summarization dataset using the two-step pessimistic learning framework.

Recall that the two-step learning framework consists of a stage of standard reward modeling process (RM) followed by a pessimistic reward fine-tuning process (PET), and a stage of proximal policy optimization (PPO). Note that its difference from the traditional RM+PPO two-step learning framework is an additional PET training process. For each training process, we set the number of training steps to be sufficient for the training to converge. For standard RM, the number of training steps is $90K$, and the training time is 3 GPU hours. For PET, the number of training steps is $8K$, and the training time is 3.5 GPU hours. The PET process requires many fewer training steps compared to RM because it is initialized from the reward model learned at the RM stage, which is already a high-quality reward model. In this case, the total training time required by PET is not much larger than that required by RM. At the PPO stage, the number of training steps is $90K$, and the training time is around 18 GPU hours. Therefore, the PET training takes $3.5/(3+3.5+18) = 14.3\%$ of the total training time under the learning framework, so we believe the additional computational resource required by the PET step is acceptable. More importantly, PET can make a reward model pessimistic and allows greedy policy optimization on the reward model without the risk of over-optimization, which is a significant benefit.

**Computational Complexity:** Next, we compare the computational complexity in one training step of RM and PET in detail. Both RM and PET training processes are learning a reward model. Particularly, we focus on the number of forward and backward passes on the LLMs computed during training, as this is the main factor in the computational cost.

An RM learning agent stores a reward model $R$, and a PET learning agent stores a reward model $R$ and a reference policy model $\pi$. The input in a training step is a data tuple $(x, a_1 > a_2)$ where $x$ is the prompt, $a_1$ is the chosen response, and $a_2$ is the rejected response.

At each training step, the RM agent scores the chosen and rejected responses with the reward model $R(x, a_1)$ and $R(x, a_2)$. Then the reward prediction loss is computed as $-\log(\sigma(R(x, a_1) - R(x, a_2)))$. In the end, the agent performs a gradient descent step on the reward model based on the prediction loss. In total, RM at each training step requires 2 forward passes (model inference) and 1 backward pass (gradient descent) on the model.

For the PET agent, it computes the rewards for the chosen and rejected responses $R(x, a_1), R(x, a_2)$ same as the RM agent. Let $n$ be the number of samplings during the best of n sampling process. At each training step, it generates $n$ samples from the reference policy $s_1, \ldots, s_n \sim \pi(\cdot|x)$, and then scores each sample with the reward model $R(x, s_1), \ldots, R(x, s_n)$. We denote the response with the highest score as $s^*$. Then, it generates one sample from the reference policy $s' \sim \pi(\cdot|x)$ and scores it with the reward model $R(x, s')$. In the end, the PET loss is computed as $-\log(\sigma(R(x, a_1) - R(x, a_2))) + \beta \cdot (R(x, s^*) - R(x, s'))$, where $beta$ is the pessimistic coefficient. The agent performs a gradient descent step on the reward model based on the PET loss. In total, one training step in PET requires $(2 + 2n + 2) = 2n + 4$ forwards passes and 1 backward pass on the models. The cost of the full training stage can be characterized as:

- RM: $2 \cdot N_1$ for forward and $N_1$ backward where $N_1$ is the total number of training steps.

- PET: $(2n + 4) \cdot N_2$ for forward and $N_2$ backward where $N_2$ is the total number of training steps

Note that in practice, the PET training requires many fewer training steps compared to the RM training. In our experiments, we have $N_1 = 90K$, $N_2 = 8K$, and $n = 64$. In this setup, the whole PET training process takes $876K$ more forward passes but $82K$ fewer backward passes on the reward model compared to the RM training process. Since the backward pass on a model is more computationally expensive than a forward pass, the total training time of PET (3.5 GPU hours) is only slightly larger than the total training time of RM (3 GPU hours) in our experiments.

## A.5 RELATED WORK

### A.5.1 RLHF BASED ON EXPLICIT REWARD MODELING

It is typical in RLHF to train an explicit reward model first and then optimize a policy on the learned reward model. A popular choice for policy optimization is the special rejection sampling technique known as best-of-n sampling (BoN) (Bai et al., 2022b; Touvron et al., 2023). Theoretical understandings of the properties of the BoN process have been developed to explain the efficiency of the method (Beirami et al., 2024; Huang et al., 2025). Empirically, different variants of BoN have been proposed to improve the learning efficiency and robustness against reward hacking (Huang et al., 2025; Jinnai et al., 2024; Liu et al., 2023; Xiong et al., 2025; Khaki et al., 2024). Another popular choice for policy optimization is to use the PPO algorithm (Schulman et al., 2017). In this case, it is prevalent in current methods to add a KL regularization in the policy optimization goal to avoid reward hacking (Stiennon et al., 2020; Bai et al., 2022a; Ouyang et al., 2022; Christiano et al., 2017). To improve the learning efficiency of KL-PPO, methods have been proposed to improve the quality of the reward model (Liu et al., 2024a; Sun et al., 2025; Shen et al., 2024; Ramé et al., 2024; Coste et al., 2023) or further constrain the agent from uncovered policies (Dai et al., 2025)

### A.5.2 RLHF BASED ON DIRECT PREFERENCE ALIGNMENT

Another typical RLHF technique skips the step of explicit reward modeling through direct preference alignment, also known as direct preference optimization (DPO) (Rafailov et al., 2023). In DPO, a reward model is implicitly represented by a policy model, so the agent can directly train a policy model to minimize the prediction loss. Numerous direct preference alignment techniques have been proposed to achieve a higher learning efficiency (Azar et al., 2024; Xiong et al., 2023b; Tang et al., 2024; Ji et al., 2024a; Liang et al., 2024; Rosset et al., 2024). Provably robust algorithms against reward hacking based on DPO have also been developed (Fisch et al., 2024; Zhang et al., 2024; Xie et al., 2024; Liu et al., 2024b; Cen et al., 2024; Ji et al., 2024b). Note that KL regularization plays a

critical role in DPO, as the regularization term makes it possible to represent a reward model by the optimal policy on it under the regularization.

### A.5.3 RLHF WITH ONLINE PREFERENCE COLLECTION

This work focuses on RLHF in the offline setting, where the learning agent only has access to an offline preference dataset. Another important setting of RLHF is the online setting, where an agent can collect preference data online. The online setting can break the limit of the offline setting in that the agent can control the data distribution to cover high-quality policies actively. The online setting is also more expensive than the offline setting, as it needs to collect customized data during training. Theoretical studies have developed efficient online exploration algorithms to improve the quality of online data collection (Qi et al., 2025; Wu & Sun, 2023; Zhao et al., 2025). Practical methods and learning frameworks have also been developed and have achieved promising empirical results (Dong et al., 2024; Xiong et al., 2023a; Bai et al., 2022a; Touvron et al., 2023).

**Use of Large Language Models:** This work only uses LLMs to polish writing and correct grammatical mistakes.

