# OpenReview forum: "Pessimistic Reward Modeling in RLHF against Reward Hacking"
_ICLR.cc/2026/Conference — Submitted to ICLR 2026_

### Official Review · Reviewer_GwNQ · 2025-10-29

**Soundness:** 2
**Presentation:** 2
**Contribution:** 2
**Rating:** 4
**Confidence:** 2

**Summary:**

This paper proposes PET,  a method to fine-tune a pessimistic reward model that can prevent reward hacking in offline RLHF without using KL regularization. The main idea is to adversarially train a reward model against a BoN policy. More specifically, the policy (actor) seeks to maximize the pessimistic reward. The reward model (critic) minimizes its score on that policy while fitting the preference dataset.

**Strengths:**

- PET-PPO seems to outperform or match SOTA RLHF methods across the tested benchmarks.
- The authors also provide theoretical guarantees on the performance of the pessimistic reward fine-tuned by PET.
- The observation that PET mitigates length bias is a nice practical insight.

**Weaknesses:**

**Confusion around the role of KL divergence:**
It is unclear whether the authors view KL regularization as too pessimistic or not pessimistic enough.
The text alternates between claiming that KL induces “over-pessimism” and that it is “not pessimistic enough” to prevent reward hacking.
This needs a clear explanation and empirical support (when and why KL over- or under-compensates). It is unclear why making the reward models pessimistic without using the KL divergence is an important problem to solve.

**Ambiguity in labeling**:
Unclear what the difference is between tables 3 and 4.

**Algorithm 2 does not seem like a substantial contribution**
It mainly wraps PET into the standard two-step RLHF framework.The contribution lies in Algorithm 1 (PET).

**Trustworthiness and pessimistic claims not substantiated:**
- The statement “This empirically verifies that the pessimistic reward fine-tuned by PET is trustworthy” is not justified. The evidence given (strong downstream performance) is indirect.
- It is unclear whether the PET reward models are actually pessimistic. There is no quantitative measure showing PET’s rewards are systematically lower or more conservative than the proxy’s. They can produce policies with a larger KL divergence. But the authors should compare predicted rewards from the proxy and PET models for the same responses.

**Limited empirical scope and lack of statistical analysis**
- Experiments use only two small datasets (TL;DR, IMDB) and a single model (Pythia-1B) with LLM-based evaluation (Qwen-2.5-32B).
- The experiments appear to use only a single random seed. No confidence intervals or variance estimates are provided, making it difficult to judge whether improvements are statistically meaningful or within noise. Especially given the small magnitude of some gains.

**Questions:**

- How many random seeds were used for each experiment?
- The authors note that policies learned with the PET reward can have high real performance while having a large KL divergence. Why is this happening, and why is it beneficial to have a large KL divergence?
- Can the authors explain the difference between Tables 3 and 4?
- The choice of β (pessimistic coefficient) is fixed at 1/β = 0.1. Why was it set this way? What happens if we change that value?
- Do the authors think the other baselines would work better if they used a higher value for n? Since it was noted several times that n was set to 64 because of fewer computational resources?

---

> ### Author Response · Authors · 2025-11-29
>
> **Q1:** It is unclear whether the authors view KL regularization as too pessimistic or not pessimistic enough. The text alternates between claiming that KL induces “over-pessimism” and that it is “not pessimistic enough” to prevent reward hacking. This needs a clear explanation and empirical support (when and why KL over- or under-compensates). It is unclear why making the reward models pessimistic without using the KL divergence is an important problem to solve.
>
> **A1:** We have explained in the paper that KL regularization is only an intuitive approach to control the degree of pessimism. Therefore, it can be over-pessimistic for some policies while under-pessimistic for other policies. As a result, it is important to develop a more efficient approach that can better control the degree of pessimism compared to intuitive approaches like KL regularization. Traditional approaches based on KL penalties exclude policies with high KL divergence, including the ones with high-quality. Without relying on penalties for KL divergence, our method has the ability to learn high-quality policies with high KL divergence, which are empirically verified in Table 2.
>
> Our empirical evaluation in Table 2 shows that our method learns high-quality policies with high KL divergence.
>
> **Q2:** Unclear what the difference is between tables 3 and 4.
>
> **A2:** We explain the tables 3 and 4 clearly in both text and captions. Table 3 shows the performance of a policy compared to baseline human responses. Table 4 shows the performance of a policy compared to policies learned by different algorithms.
>
> **Q3:** It mainly wraps PET into the standard two-step RLHF framework. The contribution lies in Algorithm 1 (PET).
>
> **A3:** PET is an algorithm to learn a pessimistic reward model instead of a policy model, while the goal of an RLHF algorithm is to learn a policy model. Therefore, it is vital to combine PET with other policy optimization methods to learn a high-quality policy model from a pessimistic reward model. Our contribution lies in both PET (to learn a pessimistic reward model) and its combination with the two-step learning framework (to learn a high-quality policy model).
>
>
> **Q4:** The statement “This empirically verifies that the pessimistic reward fine-tuned by PET is trustworthy” is not justified. The evidence given (strong downstream performance) is indirect.
>
> **A4:** We clarify that a high KL divergence indicates a high degree of policy optimization. Therefore, since PET learns a high-quality policy with high KL divergence, it implies that unconstrained policy optimization on the pessimistic reward model learned by PET is safe. Therefore, we claim that the pessimistic reward fine-tuned by PET is trustworthy.
>
> **Q5:** It is unclear whether the PET reward models are actually pessimistic. There is no quantitative measure showing PET’s rewards are systematically lower or more conservative than the proxy’s. They can produce policies with a larger KL divergence. But the authors should compare predicted rewards from the proxy and PET models for the same responses.
>
> **A5:** The PET training process is initialized from a proxy reward. During training, the reward for the responses generated through the BoN process consistently decreases, which is already an evidence for the learned reward model being more and more pessimistic. Another potentially fair comparison is to normalize the reward predictions of a reward model on a set of responses and compare the normalized reward predictions between different reward models. When we compare the pessimistic reward model and the proxy reward model on the IMDB dataset in this way, we find that in most cases, the pessimistic reward model assigns a lower reward, which also supports the claim that it is more conservative than the proxy reward.

---

> ### Author Response · Authors · 2025-11-29
>
> **Q6:** Experiments use only two small datasets (TL;DR, IMDB) and a single model (Pythia-1B) with LLM-based evaluation (Qwen-2.5-32B).
>
> **A6:** See our response A1 to reviewer V5bP.
>
> **Q7:** The experiments appear to use only a single random seed. No confidence intervals or variance estimates are provided, making it difficult to judge whether improvements are statistically meaningful or within noise. Especially given the small magnitude of some gains.
>
> **A7:** Due to the budget of computational resources, we only have one random seed, which is also common in many RLHF literature [1,2]. We report all the training configurations and the random seed to ensure reproducibility.
>
> **Q8:** The authors note that policies learned with the PET reward can have high real performance while having a large KL divergence. Why is this happening, and why is it beneficial to have a large KL divergence?
>
> **A8:** The large KL divergence of the learned policy is the consequence of removing the KL constraint during policy optimization. There could be high-quality policies that have high KL divergence, but they will be excluded by the KL penalty widely used in current RLHF approaches. Our work is the first to break this limit so that it is possible to find more high-quality policies from the same offline preference dataset.
>
> **Q9:** Do the authors think the other baselines would work better if they used a higher value for n? Since it was noted several times that n was set to 64 because of fewer computational resources?
> **A9:** Among the baseline methods we consider, only the bon algorithm itself relies on bon. Its performance may not increase when using a higher value for n because of reward hacking. A high value of n can mislead the algorithm to generate low-quality responses overestimated by the proxy reward model. This phenomenon has been observed and discussed in related literature, such as Figure 11 in [3].
>
> [1]: Liu, Zhihan, et al. "Provably mitigating overoptimization in rlhf: Your sft loss is implicitly an adversarial regularizer." Advances in Neural Information Processing Systems 37 (2024): 138663-138697.
>
>
> [2]: Rafailov, Rafael, et al. "Direct preference optimization: Your language model is secretly a reward model." Advances in neural information processing systems 36 (2023): 53728-53741.
>
>
> [3]: Huang, Audrey, et al. "Is best-of-n the best of them? coverage, scaling, and optimality in inference-time alignment." arXiv preprint arXiv:2503.21878 (2025).

---

### Official Review · Reviewer_V5bP · 2025-11-01

**Soundness:** 3
**Presentation:** 2
**Contribution:** 2
**Rating:** 4
**Confidence:** 3

**Summary:**

This paper introduces PET (Pessimistic Reward Fine-Tuning), a method for training pessimistic reward models in Reinforcement Learning from Human Feedback (RLHF) to prevent reward hacking without using KL regularization. Instead of constraining policy updates through KL penalties, PET adversarially fine-tunes the reward model using a Best-of-N sampling process so that it remains faithful to human preference data while assigning lower scores to over-optimized outputs. This produces a pessimistic reward model that enables greedy policy optimization without overfitting or reward exploitation. Experiments on summarization and sentiment-generation tasks show that PET-trained policies achieve higher real-world performance, reduced length bias, and robustness to reward hacking, while maintaining efficiency and outperforming several state-of-the-art RLHF baselines.

**Strengths:**

The paper introduces PET, a novel pessimistic reward fine-tuning method that mitigates reward hacking without relying on KL regularization. Combining adversarial training with BoN sampling indeed opens up new opportunities.
And the presentation is clearly formalized with both theoretical guarantees and experimental evaluations provided.

**Weaknesses:**

1. The biggest weakness is limited experiments. It would be more convincing if more recent models were tested, like Qwen, llama, and GPT (they do support fine-tuning via API). The current task is too simple to conclude that PET helps the training without using KL.

2. The adversarial training framework reminds me of self-play and also [PAIRED](https://www.google.com/url?sa=t&source=web&rct=j&opi=89978449&url=https://research.google/blog/paired-a-new-multi-agent-approach-for-adversarial-environment-generation/&ved=2ahUKEwj2u-6I1dCQAxUcEFkFHTWKA3sQFnoECBsQAQ&usg=AOvVaw2uO7Rk9lfGYI4faitR3oyu). It would be better to add some citations and discussions.

**Questions:**

1. Have you tested the method in tasks like math reasoning, coding, or hallucination benchmarks?
2. Regarding the reward model quality, could you evaluate it against the [reward bench](https://huggingface.co/spaces/allenai/reward-bench)?

---

> ### Author Response · Authors · 2025-11-29
>
> **Q1:** The biggest weakness is limited experiments. It would be more convincing if more recent models were tested, like Qwen, llama, and GPT (they do support fine-tuning via API). The current task is too simple to conclude that PET helps the training without using KL.
>
>
> **A1:** We clarify that the tasks we consider are standard benchmarks for RLHF algorithms [1]. We agree that a more comprehensive empirical evaluation can make the contribution of this work more significant. Therefore, we plan to add new experiments to make the evaluation more comprehensive. 1. Test on the ultra-feedback dataset with the latest Qwen model. 2. Test on the summarization dataset with pythia-3b and pythia-7b models as an ablation study on the model size.
>
> **Q2:** The adversarial training framework reminds me of self-play and also PAIRED. It would be better to add some citations and discussions.
>
>
> **A2:**  Thanks for pointing out the additional related literature. We will include them in the revision.
>
> [1]: Chaudhari, Shreyas, et al. "Rlhf deciphered: A critical analysis of reinforcement learning from human feedback for llms." ACM Computing Surveys 58.2 (2025): 1-37.
>
> **Q3:** Regarding the reward model quality, could you evaluate it against the reward bench?
>
> **A3:** Reward bench is more suitable to evaluate the accuracy of a reward model, e.g, can it figure out whether one response has a higher quality than another. The main focus of our reward model is to be pessimistic, meaning it should assign low rewards to responses not covered by the current offline dataset. Therefore, we believe it is not appropriate to evaluate the pessimism of a reward model on the reward benchmark.

---

### Official Review · Reviewer_kbgN · 2025-11-02

**Soundness:** 3
**Presentation:** 3
**Contribution:** 2
**Rating:** 4
**Confidence:** 4

**Summary:**

The paper proposes PET, an adversarial reward-model fine-tuning method for offline RLHF that aims to prevent reward hacking without KL regularization. PET proposes a minimax game between a reward model and a BoN policy; leveraging a key property of BoN, the game reduces to a tractable single minimization solved by SGD. The authors then run a two-step pipeline: train the pessimistic reward with PET, and optimize a policy on it greedily. Empirically, the authors show BoN on the PET reward improves win-rates over BoN on a proxy reward; PPO without KL on the PET reward reaches competitive or top performance while having large KL, which contradicts the usual "high-KL leads to reward hacking" case. PET also reduces length bias in reward models. A finite-sample theorem guarantees that, for policies covered by data, the PET solution competes with any BoN policy.

**Strengths:**

Originality
1. The paper reformulates adversarial reward-policy training by choosing BoN as the adversary, which yields a clean reduction to SGD (as a single step update), which overcomes the instability of standard adversarial RLHF and dispenses with KL regularization during policy optimization.

Quality
1. Clear derivation, where the claims are backed with proofs.
2. Implementation details and complexity analysis are provided.

Clarity
1. The two-step diagram is very clear.

Significance
1. No-KL PPO on PET outperforms KL-PPO on PET and avoids reward hacking despite high KL which is a notable empirical result.
2. Length bias is substantially reduced by PET (scatter analysis).

**Weaknesses:**

1. Coverage assumption and BoN-specificity. The theorem requires dataset coverage of candidate BoN policies; in realistic deployments, high-quality but poorly covered behaviors can matter most. PET’s guarantees do not extend there, and the method targets around BoN (though PPO is used post-PET), which may constrain generality.
2. Parameter selection. PET introduces a pessimistic coefficient $\beta$ while in the experiments the authors only follow specific setup in previous paper. Does this selection follow the best practice suggested by Theorem 3.1? Does this selection allows convergence to a good policy for most usecases? Any practical suggestions?
3. Theory. The theorem and its proof seem a direct implication from Liu et al. (2024b) and  Zhan et al., (2023). Can the authors clarify if any theoretical contributions they may have?
4. Confound between KL and training setup. The claim that PPO without KL works well could be confounded by training length/step budget, early stopping, or implicit regularizers (e.g., entropy bonuses, clip ranges).
5. BoN. Does PET generalize well, i.e., does it work for non-BoN inference-time optimizers?

**Questions:**

See above.

---

> ### Author Response · Authors · 2025-11-29
>
> **Q1:** The theorem requires dataset coverage of candidate BoN policies; in realistic deployments, high-quality but poorly covered behaviors can matter most. PET’s guarantees do not extend there, and the method targets around BoN (though PPO is used post-PET), which may constrain generality.
>
> **A1:** We clarify that what PET guarantees is already the gold standard one can hope for in the offline learning setting [1]. If high-quality behaviors are not covered by the dataset, then it is impractical for the learning agent to learn that those behaviors are of high quality. In fact, reward hacking is exactly the consequence of overestimating the behaviors that are not covered by the dataset.
>
> **Q2:** PET introduces a pessimistic coefficient while in the experiments, the authors only follow a specific setup in a previous paper. Does this selection follow the best practice suggested by Theorem 3.1? Does this selection allows convergence to a good policy for most use cases? Any practical suggestions?
>
> **A2:** We clarify that the pessimistic coefficient is prevalent in existing literature to control the degree of pessimism [2]. To ensure fairness in comparison, our setup of the pessimistic coefficient follows the standard setup in previous work [3], and it achieves good performance in practice.
>
> **Q3:** Theory. The theorem and its proof seem a direct implication from Liu et al. (2024b) and Zhan et al., (2023). Can the authors clarify if any theoretical contributions they may have?
>
> **A3:** We clarify that our unique theoretical contribution is to show that the adversarial training problem proposed in Eq 1 can be solved by a standard stochastic gradient descent process if we focus on the best-of-n policies. The details can be found in Appendix A1.
>
> **Q4:** Confound between KL and training setup. The claim that PPO without KL works well could be confounded by training length/step budget, early stopping, or implicit regularizers (e.g., entropy bonuses, clip ranges).
>
> **A4:** We clarify that our experiments adopt the standard setups in RLHF literature. The performance of our algorithm in our experiments does not benefit from any additional experimental tricks compared to the baseline algorithms we consider.
>
> **Q5:** Does PET generalize well, i.e., does it work for non-BoN inference-time optimizers?
>
> **A5:** Our main focus is to build an approach to learn a pessimistic reward model for RLHF. Therefore, developing a general framework that can work with all inference-time policy optimization techniques is out of the scope of this work. It could be an interesting future work to study combining the PET framework with other inference-time optimizers.
>
> [1]: Huang, Audrey, et al. "Correcting the mythos of kl-regularization: Direct alignment without overoptimization via chi-squared preference optimization." arXiv preprint arXiv:2407.13399 (2024).
>
> [2]: Wang, Zhichao, et al. "A comprehensive survey of llm alignment techniques: Rlhf, rlaif, ppo, dpo and more." arXiv preprint arXiv:2407.16216 (2024).
>
> [3]: Liu, Zhihan, et al. "Provably mitigating overoptimization in rlhf: Your sft loss is implicitly an adversarial regularizer." Advances in Neural Information Processing Systems 37 (2024): 138663-138697.

---

### Meta-Review · Area_Chair_Pkg1 · 2025-12-24

**Summary:**

The present paper considers the reinforcement learning from human feedback setting, which traditionally learns a reward model on human feedback to model-generated outputs. Using RLHF in the training of large models is prone to reward hacking, which is the key problem addressed in this work. Its main contribution is a pessimistic reward fine-tuning technique for offline RLHF that results in a model more robust to reward hacking.

**Reviewer Concerns:**

The reviewers raised several concerns, mainly centered around the sufficiency of experiments and the novelty of the theoretical contribution. Many points were also about how well-substantiated individual claims are.

**Reviewer Scores:**

Two of the reviewers are potentially AI-generated, although one of those two is more likely to have been used for language improvement rather than the actual evaluation. Based on this situation, it is hard to judge, and I evaluated the author's engagement with the reviewer's core arguments independently.

I would expect the concerns on scope and novelty threshold raised by reviewer kbgN to not be sufficiently addressed by the authors to the satisfaction of the reviewer. Reviewer V5bP requested additional experiments, which the authors promised, and a more extensive literature review. It is also doubtful whether the relatively vague answer on additional related work would have been sufficient in the eyes of the reviewer. Finally, some of the questions of reviewer GwNQ were factual in nature and answered by the authors. And while not all of the may have been answered to the reviewer's satisfaction, I would expect this reviewer to have the strongest grounds for increasing their score.

---

### Decision · Program_Chairs · 2026-01-26

Reject